# OpenReview forum: "GRACE: Discriminator-Guided Chain-of-Thought Reasoning"
_EMNLP/2023/Conference — EMNLP 2023 Findings_

### Official Review · Reviewer_GMbU · 2023-07-20

**Soundness:** 4

**Excitement:**

3: Ambivalent: It has merits (e.g., it reports state-of-the-art results, the idea is nice), but there are key weaknesses (e.g., it describes incremental work), and it can significantly benefit from another round of revision. However, I won't object to accepting it if my co-reviewers champion it.

**Paper Topic And Main Contributions:**

The paper proposes the addition of a discriminator that judges whether each step in a CoT is correct, whenever that step is generated. Using the discriminator that outputs a score, the most-likely-correct step can be sampled dynamically in a CoT. Performance on various symbolic reasoning datasets show the efficacy of said method.

**Questions For The Authors:**

Minor things:
- L154-182: How does the formulation help with the story? The summarized points that (i) high LM likelihood and (ii) correctness is preferred seem evident. Please provide more intuitive explanation as to why the discussion of mathematical formulation is necessary.
- Sec2.2-Alignment: The proposed approach of alignment makes intuitive sense (though again I would argue the formulation does not add much to the overall story). However, it is imperative to show that the steps aligned in this way are empirically correct, using some sort of correlation with human judgement, even on a small subset of data.

**Reasons To Accept:**

1. The idea of using the discriminator is motivated by the shortcoming of existing voting methods that require generating the whole CoT, which costs more resource.
2. The proposed method yields clear empirical gain in various datasets.
3. The discussions in Sec 4 and 5 are fair and sound.
4. The paper is clearly and convincingly presented.

**Reasons To Reject:**

1. The critical component of the proposed approach is the discriminator, which needs training data. While aligned steps can be obtained automatically from the training set of a dataset, this effectively makes the entire approach supervised, even though the final model that makes inference (e.g., your LLaMA7B) is in-context / unsupervised. Such a condition might not be met in many other tasks and datasets without a sizeable training set (e.g., your 80K incorrect solutions). In this case, it would strengthen the paper to show either of the following, if possible:
  - if the discriminator can transfer to other tasks and domains without further training
  - if the discriminator can be effectively trained with very little data
In the same vein, I would mention that at L400 you already show that the discriminator can be small. That is a very important insight.
2. The paper would benefit from stronger motivation (other than empirical performace gain) of why voting per step is preferred to voting per CoT (previous work). This is fleetingly discussed in L72-77 which is insufficient.
  - your approach also relies on temperature sampling (Sec2.2-Negative Sampling); in this case, how would it help with "sampling likely but incorrect solutions"?
  - why exactly is "control over the solution sampling process" beneficial in real-life applications? (my hypothesis is that it decreases cost, but it would be nice to thoroughly discuss this)

**Reproducibility:**

5: Could easily reproduce the results.

**Reviewer Confidence:**

5: Positive that my evaluation is correct. I read the paper very carefully and I am very familiar with related work.

**Typos Grammar Style And Presentation Improvements:**

- L350: remove "it"

---

> ### Author Rebuttal · Authors · 2023-08-28
>
> Thank you for your very insightful comments. We are glad that you have found our paper clearly and convincingly presented.
>
>
> > While aligned steps can be obtained automatically from the training set of a dataset, this effectively makes the entire approach supervised.
>
>
> We agree with you that our approach assumes access to gold reasoning chains for the alignment. In cases where such gold chains do not exist (we only have the question and the final answer), one route is to prompt an LLM to generate the step-by-step solutions as we have done with some of the datasets as mentioned in L302-308.
>
>
> >  It would strengthen the paper to show either of the following, if possible:
> >  if the discriminator can transfer to other tasks and domains without further training
>
>
> Thank you for this valuable suggestion. In **Table II** below, we report the results of using GSM8K discriminator during inference over SVAMP and MultiArith with LLaMA-7B. GRACE exhibits a relative performance drop of only 8% for GSM8K --> SVAMP and 7% for GSM8K --> MultiArith, and still substantially outperforms greedy and LM-only scoring. When transferring the discriminator from SVAMP to MultiArith, we note a more significant relative performance drop (26.6%). This might be because SVAMP has fewer training questions (432) compared to GSM8K (6.4K), and SVAMP questions tend to involve simpler reasoning, resulting in a discriminator with worse generalization (although its performance still outperforms the two baselines). We will include these results in the paper revision.
>
>
>
>
> |                   | Acc. (SVAMP) | % Drop |                   | Acc. (MultiArith)  | % Drop |
> |-------------------|------|------|-------------------|-------|------|
> | Greedy            | 32.8 |-| Greedy            | 54.0  |
> | LM-only scoring   | 39.6 |-| LM-only scoring   | 57.9  |
> |-------------------|------|------|-------------------|----|
> | GRACE  | 49.7 |-| GRACE | 84.9  | -
> | GRACE (trained on GSM8K) | 45.5 |8%| GRACE (trained on GSM8K)  | 78.5  | 7%
> | -- | -- | --| GRACE (trained on SVAMP)  | 62.3  | 26.6%
>
>
> **Table II: LLaMA-7B performance when transferring the discriminator across different datasets. Results are averaged over 3 runs.**
>
>
>
>
> > The discriminator can be effectively trained with very little data In the same vein
>
>
> Thanks again for the insightful suggestion. First, we note that the discriminator was trained on a relatively small number of examples for many of the tasks we experiment with. For instance, MultiArith has only 289 gold trajectories in total and Coin Flip has only 245 (see Table 7 in the Appendix for dataset statistics). However, we certainly agree on the benefit of analyzing the performance of the discriminator as a function of the training data size to study the prospect of a few-shot discriminator. Due to the limited time of the rebuttal, we will include this analysis in the future revision.
>
>
> > The paper would benefit from stronger motivation (other than empirical performance gain) of why voting per step is preferred to voting per CoT (previous work). This is fleetingly discussed in L72-77 which is insufficient.
>
>
> Thank you for this suggestion. In addition to the discussion in L72-77, we also return to this notion of "control over the process" in L463-466 when discussing the sampling efficiency of our approach compared to vanilla self-consistency. We will make sure to highlight this motivation more strongly in the revision.
>
>
> One interesting analogy is comparing chain vs. step voting to sparse vs. dense rewards in RL. Guiding the solution at the step level is more like providing rewards at the intermediate steps rather than a delayed reward signal at the end of the episode. Clearly, one main benefit here is efficiency i.e., the requirement for fewer samples to reach a correct answer, which we will expand more on below when addressing another point.
>
>
> > Your approach also relies on temperature sampling (Sec2.2-Negative Sampling); in this case, how would it help with "sampling likely but incorrect solutions"?
>
>
> The main utility of temperature sampling is to simulate the LM reasoning with the goal of exploring the model's reasoning error space. While it may be more beneficial to focus on the higher likely errors by using a small temperature constant, this comes at the expense of the diversity of the errors. One can think of this as an exploration/exploitation tradeoff. While we have not done extensive tuning of the sampling temperature parameter, we have found that higher temperatures i.e., allowing more exploration of the error space yield a larger and more diverse pool of negatives than lower temperatures.
>
> > Why exactly is "control over the solution sampling process" beneficial in real-life applications? (my hypothesis is that it decreases cost, but it would be nice to thoroughly discuss this)
>
>
> Your intuition is indeed correct. To illustrate how cost may be reduced when using GRACE as compared to self-consistency/verifier, let's assume that on average each solution has 5 steps and that each step consists of 20 tokens. Standard voting per CoT with the number of samples = 40 will require sampling of 20 x 5 x 40 = 4000 tokens in total per question. Assuming our step-wise voting approach will sample and score 10 candidate steps at a time, we will only need to sample 20 x 5 x 10 = 1000 tokens per question i.e., 4x fewer tokens than needed for self-consistency, while often achieving higher performance. We will expand on this in the revision following your suggestion.
>
>
> > L154-182: How does the formulation help with the story? The summarized points that (i) high LM likelihood and (ii) correctness is preferred seem evident. Please provide a more intuitive explanation as to why the discussion of mathematical formulation is necessary.
>
>
> The main motivation for providing the formulation is to connect our work with prior work on the controllable generation, namely FUDGE, so the reader can easily contrast our step-wise formulation with the token-wise formulation of FUDGE. We agree that the summarized points could be intuitive to some readers.
>
>
> > Sec2.2-Alignment: The proposed approach of alignment makes intuitive sense (though again I would argue the formulation does not add much to the overall story). However, it is imperative to show that the steps aligned in this way are empirically correct, using some sort of correlation with human judgement, even on a small subset of data.
>
>
> Thank you for this suggestion. Indeed we performed a manual inspection of the alignment outputs over a small set of solutions to validate and tune the alignment algorithm. In addition, by evaluating the downstream reasoning performance of the discriminator, we are extrinsically evaluating the quality of the produced alignments.

---

### Official Review · Reviewer_2yfe · 2023-07-28

**Soundness:** 4

**Excitement:**

4: Strong: This paper deepens the understanding of some phenomenon or lowers the barriers to an existing research direction.

**Paper Topic And Main Contributions:**

This paper presents a step-wise decoding method for multi-step reasoning. Language models tend to assign high probabilities scores to incorrect steps, and sampling multiple steps does not fully solve this problem. To this end, this paper proposes to use a discriminator (a small LM) to aid a base LM during decoding. The discriminator is trained with negative sampling and a contrastive loss, and uses an alignment algorithm for steps of different length. During decoding, the base LM generates each step by top-p sampling, whereas the discriminator determines whether the step is correct. On 4 math and 2 symbolic reasoning tasks, this method outperforms baselines like greedy decoding and Self-Consistency, and also generates more correct reasoning steps. Analyses justify some design choices of the discriminator.

**Questions For The Authors:**

A. For the discriminator, how to identify intermediate variables and compare them?

B. In eq. (6), have you tried decoding algorithms other than nucleus sampling for the LM?


**Reasons To Accept:**

1. The idea of step-wise decoding instead of token-wise decoding is novel and sensible. Modifying the Needleman-Wunsch (NW) algorithm to align steps of different length is novel and has practical benefits.
2. The method performs well, and main claims are supported by experiments and analysis.
3. This paper is well-written, and provides sufficient details to clarify key concepts like algorithm 1.


**Reasons To Reject:**

1. The proposed method is limited to models that provide logits. It is not straightforward to apply to LLMs like ChatGPT.
2. This paper does not thoroughly cover multi-step reasoning tasks. It would be better to include commonsense reasoning tasks.



**Reproducibility:**

4: Could mostly reproduce the results, but there may be some variation because of sample variance or minor variations in their interpretation of the protocol or method.

**Reviewer Confidence:**

3: Pretty sure, but there's a chance I missed something. Although I have a good feel for this area in general, I did not carefully check the paper's details, e.g., the math, experimental design, or novelty.

**Typos Grammar Style And Presentation Improvements:**

Presentation: I found figure 4 to be very helpful to understand the alignment. It may be better to present it in the main text instead of the appendix.

---

> ### Author Rebuttal · Authors · 2023-08-28
>
> Thank you so much for your comments. We are glad that you found our approach novel, sensible and to have practical benefits.
>
> > The proposed method is limited to models that provide logits. It is not straightforward to apply to LLMs like ChatGPT.
>
> As we mention in L185-187 and 484-486, we ideally want to leverage the LM's reasoning abilities when computing a candidate step score by interpolating the LM step probability and the discriminator score in Equation 6. However, in the case of lack of access to the model logits, our approach can be directly applied by setting $\beta = 1.0$ in Equation 6. **Table 1** below shows the performance of 6-shot GPT-3.5-turbo over GSM8K test set. We use the same 6-shot prompt used with LLaMA in our paper (shown in Table 8 in the Appendix). GRACE sampling parameters are T=0.9, top_p=0.85, and J (number of candidate steps) = 10. Self-consistency is done with N=20 samples and with T=0.7. Even without access to the logits, GRACE still yields substantial accuracy improvement (11.3% points) over self-consistency. We will include these results in the next paper revision.
>
> | Method | Accuracy|
> |--------|--------|
> | Greedy decoding  | 53.8 |
> | Self-consistency | 55.2 |
> | GRACE | 66.5 |
>
> **Table I: Results on GSM8K with GPT-3.5-turbo.**
>
> > This paper does not thoroughly cover multi-step reasoning tasks. It would be better to include commonsense reasoning tasks.
>
> We validate the effectiveness of our approach over 6 different multi-step reasoning tasks involving both math and symbolic reasoning, which we believe is sufficient to demonstrate the effectiveness of our method. We realize there are other suites of tasks such as commonsense reasoning and planning where our approach is indeed applicable and we will certainly consider evaluating our method over these in future work.
>
> > A. For the discriminator, how to identify intermediate variables and compare them?
>
> We use simple regex matching to find the number that follows the `=` operator in a given step, i.e., the output of the formula. If no such formula exists, we return the last occurring number in the given step. Then we compare that number to the number in the aligned ground-truth step.
>
>
> > B. In eq. (6), have you tried decoding algorithms other than nucleus sampling for the LM?
>
> We have experimented with random sampling (top_p=1.0) and beam search but found nucleus sampling to often outperform both. One main issue we found with beam search is the lack of diversity in the sampled candidate steps i.e., most steps would differ in the surface form rather than in the reasoning involved. We leave exploring more advanced step sampling methods such as diverse beam search to future work.
>
> > Presentation: I found figure 4 to be very helpful to understand the alignment. It may be better to present it in the main text instead of the appendix.
>
> Thank you for pointing this out and we are glad you found Figure 4 helpful. We will move it to the main paper with the extra space we get in case of acceptance.

---

### Official Review · Reviewer_BkE7 · 2023-07-31

**Soundness:** 3

**Excitement:**

3: Ambivalent: It has merits (e.g., it reports state-of-the-art results, the idea is nice), but there are key weaknesses (e.g., it describes incremental work), and it can significantly benefit from another round of revision. However, I won't object to accepting it if my co-reviewers champion it.

**Paper Topic And Main Contributions:**

The paper aims to improve the correctness of the intermediate reasoning steps of LMs in order to improve their task performance. The paper proposes a correctness discriminator that steers the language generation process towards more correct steps. To learn such a discriminator, the work first samples a set of incorrect solutions and then uses dynamic programming to obtain the step-wise labels for each solution. With the training data, the discriminator is trained to maximize the difference between correct and incorrect steps. During inference, LM is augmented with the discriminator to select the most correct step from a set of sampled steps as the next step repeatedly. Experiments show the effectiveness of the proposed method in improving downstream performance across both mathematical and symbolic reasoning tasks.

**Questions For The Authors:**

1.	How do you make sure the negative samples are indeed incorrect? Even with the correct reasoning steps, LMs can still provide a wrong answer.

**Reasons To Accept:**

1.	The work comes up with a method to obtain positive and negative reasoning steps automatically, which are useful for learning the discriminator.
2.	The proposed method is shown to be effective in a range of tasks.

**Reasons To Reject:**

1.	The proposed method still relies on human-annotated reasoning steps to provide positive examples. With such positive examples, perhaps one easier way to obtain the negative examples could be perturbation. It is not sure how necessary the alignment process is.
2.	The work still needs to use sampling to obtain candidate steps, which does not guarantee at least one good step exists. It is helpful to have some qualitative analysis to show the discriminator selects good steps with some examples.
3.	Given that LMs are becoming larger (over 100M), it is not sure whether the proposed method is still necessary. It is good to also conduct experiments with LMs like GPT-3.

**Reproducibility:**

4: Could mostly reproduce the results, but there may be some variation because of sample variance or minor variations in their interpretation of the protocol or method.

**Reviewer Confidence:**

4: Quite sure. I tried to check the important points carefully. It's unlikely, though conceivable, that I missed something that should affect my ratings.

---

> ### Author Rebuttal · Authors · 2023-08-28
>
> We thank the reviewer for the feedback on our submission.
>
>
> > With such positive examples, perhaps one easier way to obtain the negative examples could be perturbation. It is not sure how necessary the alignment process is.
>
>
> The suggestion of using perturbation to generate negative examples is indeed interesting. However, we must acknowledge several inherent challenges with this approach. First, a *careful* set of perturbing operations needs to be designed. For instance, we may choose to perturb formulas in the case of math reasoning, but not for symbolic reasoning which necessitates a different set of perturb operations. This process, given its task-specific nature, can be intricate and time-consuming. The second problem (highlighted in L207-214) is that such perturbations are likely to result in easy negatives that will not match the distribution of reasoning errors made by the LM. For instance, if we have a perturbation operation that flips all `+` to `-` in a given correct solution and the language model does not fall into this type of error during inference (i.e., it never mistakes subtraction for addition), then training the discriminator on these perturbed chains will hardly be useful during inference. Lastly, when working with relatively long reasoning chains, the potential pool of negative examples needs to grow dramatically to cover the different combinations of perturb operations, posing an extra challenge of training the discriminator on a huge number of samples to cover the various possible reasoning error combinations.
>
>
> Our proposed sampling and alignment process sidesteps these three problems by **(i)** removing the need to design perturb operations and **(ii)** simulating mistakes the model makes at inference time via sampling, resulting in negative examples that are representative of the model reasoning errors while dramatically shrinking the space of negative samples we need to consider.
>
>
> We note that we ran early experiments based on perturbing gold chains in the context of math reasoning (GSM8K) and the performance of the resulting discriminator was *extremely* poor -- worse than greedy decoding. Manual analysis showed a lack of alignment between the LM inference errors and the discriminator training data, which inspired the proposed sampling and alignment approach. We intend to expand further on this discussion in the future revision of the paper.
>
>
>
> > The work still needs to use sampling to obtain candidate steps, which does not guarantee at least one good step exists.
>
>
> We agree with the reviewer that our alignment process assumes the existence of some correct steps in the sampled chains. However, we argue that this is not an issue as the LM can usually get some of the reasoning steps correct (for example, early steps in the reasoning chain are usually easier to get right than later steps). To be more concrete, we analyzed 25K sampled incorrect solutions from LLaMA-7B for GSM8K and found that 12% of the steps were correct i.e., they had the same intermediate variable as the corresponding ground-truth step. This is with a model that is within the 1B scale and bigger models will likely get even more steps right.
>
>
> >  It is helpful to have some qualitative analysis to show the discriminator selects good steps with some example
>
>
> Thank you for this suggestion. By looking at examples where GRACE outperformed the baselines, we observed that GRACE prevents incorrect early steps thereby keeping the reasoning on the right track. Below, we show and analyze a few of these examples with FLAN-T5-Large over GSM8K dev set. We highlight the first incorrect step picked by greedy decoding and the corresponding correct step selected by GRACE in `code`.
>
>
> #### Example #1:
> **Input Question:** At a birthday party, there are 20 red balloons and 15 green balloons. Before the party started, 3 red and 2 green balloons burst. How many balloons are left?
>
>
> **Greedy Decode:**
> `Before the party started, there were 20 + 15 = 35 red and green balloons.` So, there are 35 - 3 = 32 balloons left. Similarly, there are 15 - 2 = 13 green balloons left. Therefore, there are 32 + 13 = 45 balloons left in total. #### 45
>
>
> **GRACE:**
> `There are 20 - 3 = 17 red balloons left.` There are 15 - 2 = 13 green balloons left. There are 17 + 13 = 30 balloons left. #### 30
>
>
> #### Example #2:
> **Input Question:** Phoebe eats 1 serving and gives her dog 1 serving of peanut butter for a bedtime snack. Each jar of peanut butter has 15 servings. How many jars will she need to make sure she and her dog have enough to last for 30 days?
>
>
> **Greedy Decode:**
> Phoebe and her dog eat 1 + 1 = 2 servings a day. `If 1 jar has 15 servings, then they will need 30 / 2 = 15 jars.` #### 15
>
>
> **GRACE:**
> Phoebe and her dog eat 1 + 1 = 2 servings a day. `For 30 days, Phoebe and her dog will need 30 * 2 = 60 servings of peanut butter.` Since a jar has 15 servings, Phoebe will need 60 / 15 = 4 jars of peanut butter. #### 4
>
>
> Example #3:
>
> **Input Question:** Hank is raising money for the local homeless shelter. Hank makes $100 in a carwash and donates 90% of the proceeds to the local homeless shelter. Hank makes $80 in a bake sale and donates 75% of the proceeds to the local homeless shelter. Then he makes $50 mowing lawns and donates 100% of the proceeds to the local homeless shelter. How much money does Hank donate in total?
>
>
> **Greedy Decode:**
> First find the total amount of money Hank donates from the carwash: $100 * .9 = $90. Then find the total amount of money Hank donates from the bake sale: $80 * .75 = $60. `Then find the total amount of money Hank donates from the mowing lawns: $50 * .1 = $5.`Then add the money from each type of sale to find the total amount of money Hank donates: $90 + $60 + $5 = $155. #### 155
>
>
> **GRACE:**
> First find how much money Hank donates from the carwash: $100 * 0.9 = $90.00. Then find how much money Hank donates from the bake sale: $80 * 0.75 = $60.00. `Then find how much money Hank donates from mowing lawns: $50 * 100% = $50.00.` Then add the money from each source to find the total amount of money Hank donates: $90 + $60 + $50 = $200.00. #### 200
>
> We will expand and include this analysis in the next revision.
>
>
> > Given that LMs are becoming larger (over 100M), it is not sure whether the proposed method is still necessary. It is good to also conduct experiments with LMs like GPT-3.
>
>
> We argue that there exists a need to boost the reasoning performance of accessible and smaller LMs. Therefore, there is significant value to the research community in building and adapting different techniques to such open-source LMs. In addition, scaling LMs alone is not sufficient to solve many complex reasoning tasks, explaining the recent attention given to methods such as self-refine [1], RAP [2], tree-of-thoughts [3], etc., that are applied on top of 100B-scale LLMs with the goal of expanding their reasoning capacity. Therefore, we align our approach with this goal by mainly evaluating it on open-source LLMs such as FLAN-T5 and LLaMA with the goal of bridging the performance gap between these and large proprietary LLMs.
>
>
> To further address your concern regarding the utility of our method to other LLMs, we provide additional results of running GRACE with GPT-3.5-turbo over GSM8K. We use GPT-3.5-tubo as it is faster and cheaper than GPT-3/4. **Table I** below shows the performance of 6-shot GPT-3.5-turbo over GSM8K test set. We use the same 6-shot prompt used with LLaMA in our paper (shown in Table 8 in the Appendix). GRACE sampling parameters are T=0.9, top_p=0.85, and J (number of candidate steps) = 10. Self-consistency is done with N=20 samples and with T=0.7. Due to the limited time of the rebuttal, we directly use the discriminator checkpoint that was trained over samples from FLAN-T5-Large. We expect sampling from GPT-3.5-turbo and training another discriminator on these samples to yield even better results.
>
>
> | Method | Accuracy|
> |--------|--------|
> | Greedy decoding  | 53.8 |
> | Self-consistency | 55.2 |
> | GRACE | 66.5 |
>
>
> **Table I: Results on GSM8K with GPT-3.5-turbo.**
>
>
> As GPT-3.5-turbo's API does not provide access to the logits, we rank candidate steps here based on the discriminator score alone i.e., by setting $\beta=1.0$ in Equation 6.
>
>
> > Q: How do you make sure the negative samples are indeed incorrect? Even with the correct reasoning steps, LMs can still provide a wrong answer.
>
>
> Our alignment is computed over the intermediate steps, i.e., the steps before the model produces `#### FINAL ANSWER`. Therefore, If all the intermediate steps of a sampled solution are correct, then the steps will align well with the gold steps, and Algorithm 1 will not yield any contrastive examples, effectively discarding that solution from the negative pool since it is not really an incorrect solution. Therefore, this case is implicitly handled in Algorithm 1. We also note that in practice this happens very rarely: As the final answer is simply copied from the last step, the model usually does that right.
>
>
> We hope we have tackled all the concerns you had about our work and hope you can reconsider your evaluation. If there are any other issues, please let us know and we will be happy to resolve them in the revision.
>
>
> **References:**
>
> [1] Madaan, Aman, et al. "Self-refine: Iterative refinement with self-feedback." arXiv preprint arXiv:2303.17651 (2023).
>
> [2] Hao, Shibo, et al. "Reasoning with language model is planning with world model." arXiv preprint arXiv:2305.14992 (2023).
>
> [3] Yao, Shunyu, et al. "Tree of thoughts: Deliberate problem solving with large language models." arXiv preprint arXiv:2305.10601 (2023).

---

### Meta-Review · Area_Chair_Jfpk · 2023-09-18

**Recommendation:** 4

**Metareview:**

The manuscript introduces a novel guided decoding during generation using a discriminator. While the idea of using a discriminator in generation is not novel (example Kumar et al 2023), this is a promising research area and how the discriminator should incorporated is still an open question which this work try to address. The propose method GRACE achieves strong results, on multiple test sets compared to other decoding approaches. However, as highlighted by authors, the discriminator is run after each token generated. For example Kumar et al 2023 to avoid the significant over head of running a discriminator they leverage the encoding from the original LLM. Since this will impact feasibility of work the manuscript would have benefitted from a dedicated section.


Kumar et al 2023, "Controlled Text Generation with Hidden Representation Transformations"

---

### Decision · Program_Chairs · 2023-10-07

**Decision:**

Accept-Findings

**Comment:**

The manuscript introduces a novel guided decoding during generation using a discriminator. While the idea of using a discriminator in generation is not novel (example Kumar et al 2023), this is a promising research area and how the discriminator should incorporated is still an open question which this work try to address. The propose method GRACE achieves strong results, on multiple test sets compared to other decoding approaches. However, as highlighted by authors, the discriminator is run after each token generated. For example Kumar et al 2023 to avoid the significant over head of running a discriminator they leverage the encoding from the original LLM. Since this will impact feasibility of work the manuscript would have benefitted from a dedicated section.


Kumar et al 2023, "Controlled Text Generation with Hidden Representation Transformations"